# The Pain Divide: a cross-sectional analysis of chronic pain prevalence, pain intensity and opioid utilisation in England

Adam Todd,[1,2,3] Nasima Akhter,[2,4] Joanne-Marie Cairns,[1,2,5] Adetayo Kasim,[2,4] Nick Walton,[1,2,3] Amanda Ellison,[4,6] Paul Chazot,[4,7] Sam Eldabe,[1,8] Clare Bambra[1,2]

For numbered affiliations see end of article.

Correspondence to
Dr Adam Todd;
adam.todd@newcastle.ac.uk

## ABSTRACT

**Objectives** Our central research question was, in England, are geographical inequalities in opioid use driven by health need (pain)? To answer this question, our study examined: (1) if there are regional inequalities in rates of chronic pain prevalence, pain intensity and opioid utilisation in England; (2) if opioid use and chronic pain are associated after adjusting for individual-level and area-level confounders.

**Design** Cross-sectional study design using data from the Health Survey for England 2011.

**Setting** England.

**Primary and secondary outcome measures** Chronic pain prevalence, pain intensity and opioid utilisation.

**Participants** Participant data relating to chronic pain prevalence, pain intensity and opioid usage data were obtained at local authority level from the Health Survey for England 2011; in total, 5711 respondents were included in our analysis.

**Methods** Regional and local authority data were mapped, and a generalised linear model was then used to explore the relationships between the data. The model was adjusted to account for area-level and individual-level variables.

**Results** There were geographical variations in chronic pain prevalence, pain intensity and opioid utilisation across the English regions—with evidence of a 'pain divide' between the North and the South, whereby people in the North of England more likely to have 'severely limiting' or 'moderately limiting' chronic pain. The intensity of chronic pain was significantly and positively associated with the use of opioid analgesics.

**Conclusions** There are geographical differences in chronic pain prevalence, pain intensity and opioid utilisation across England—with evidence of a 'pain divide'. Given the public health concerns associated with the long-term use of opioid analgesics—and their questionable activity in the management of chronic pain— more guidance is needed to support prescribers in the management of chronic pain, so the initiation of opioids can be avoided.

## INTRODUCTION

Chronic pain is a worldwide problem, and the burden it places on our society is increasing: in the USA, the annual cost of chronic

### Strengths and limitations of this study

► This study is unique in that we explored the association of opioid utilisation and chronic pain.
► We adjusted for individual (eg, age and sex) and area-level confounders (eg, social deprivation) in our model.
► We did not distinguish between weak and strong opioid in our analysis, nor did we consider dose of opioid.

pain—through direct and indirect effects— is estimated to exceed US$500 billion, while in the UK estimates suggest it costs around £12 billion per year to the economy.[1 2] To manage the symptoms associated with chronic pain, some treatment strategies rely on the use of opioid analgesics, although there are very few studies to support their long-term effectiveness.[3–5] In addition, prolonged use of opioids can also have adverse consequences; this can include sleep disturbances, endocrine disorders, reduced immune function and increased pain through opioid-induced hyperalgesia.[6–10]

Despite these well-acknowledged shortcomings, the prescribing of opioid analgesics continues to increase at a significant rate.[11 12] Indeed, figures from the UK show that, in 2014, there were around 23 million prescriptions written for opioid analgesics, at a cost of around £322 million.[13] Given this increased use, (and the well-established problems associated with efficacy, tolerance and adverse effects) the inappropriate prescribing—and misuse—of opioid analgesics is becoming a significant public health concern.[14] This problem is also mirrored in other countries, such as the USA, where the death rate from opioid misuse has, in the last 15 years, quadrupled—giving rise to the so-called 'opioid epidemic'.[15]

In England, there is significant geographical variation in opioid prescribing—with more people in the North of England prescribed opioids—at a greater cost—compared with the rest of England. For example, the North of England (population of 15 million) accounts for approximately 33% of the total costs of analgesics, compared with London (population of 8.2 million), that accounts for only around 8%.[12] It is not clear, however, if this variation is related to 'inappropriate prescribing' or the varying health need of the population (ie, more people in the North of England have pain, hence the prescribing of opioids is higher). It is well documented, though, that mortality and morbidity rates are higher in the North of England, particularly in the North East region compared with the rest of England: an observation known as the North South health divide.[16]

Northern England (commonly defined as the North East, North West and Yorkshire and Humber regions) has persistently had higher all-cause mortality rates than the South of England, with people in the North consistently found to be less healthy than those in the South—across all social classes and among men and women.[17] Since 1965, this has amounted to 1.5 million excess premature deaths.[18] Further, the gap in average life expectancy between the North and the South of England is 2 years.[16] Although England is not alone in experiencing such spatial health inequalities, the divide in England is one of the largest in Europe—greater, for example, than those between the former East and West of Germany.[19] Social science suggests that the reasons for the contemporary health divide are both compositional and contextual.[16] Compositional factors include demographic factors (eg, age, sex, marital status) and socioeconomic status (eg, employment, income, education, occupation), as well as health behaviours (eg, smoking, alcohol, physical activity). In the case of pain, other compositional factors will include comorbidities such as depression or anxiety. Contextual factors include the physical (eg, air pollution or contaminated land),[20] social (eg, place-based stigma or social networks or access to services such as general practitioners)[21] and economic (eg, area-level deprivation, local job availability) environments.[22]

Given the North South health divide and public health concerns associated with the inappropriate and long-term use of opioid analgesics, it is vitally important then to explore whether the prescribing of opioid analgesics across England reflects inequalities in the health needs of the population (pain) or if there is an issue of 'inappropriate' medication prescribing or utilisation. Our central research question, therefore, was, in England, are geographical inequalities in opioid use driven by health need (pain)? To answer this question, our study examined: (1) if there are regional inequalities in rates of chronic pain prevalence, pain intensity and opioid utilisation in England; (2) if opioid use and chronic pain are associated after adjusting for individual-level and area-level confounders.

## METHODS
### Data and variables
Local authority level Health Survey for England (HSE) data were obtained from the National Centre for Social Research, which contains anonymised individual-level data and a geographical identifier (local authority district which are large administrative areas used by local government in England and have the responsibility for health and social care, education, transport and so forth). The HSE is an annual survey designed to be representative using a stratified random sample. Each year there is a focus on a particular population group, condition or disease. In 2011, one particular focus of the HSE was detailing chronic pain: as part of the wider survey, participants were asked:

► Whether they were currently troubled by pain or discomfort?
► Whether they had this pain or discomfort for more than 3 months?

If the respondent answered yes to both questions, they were categorised as experiencing chronic pain. Once it was established that participants had chronic pain, they were then asked a further three questions:

► How would you rate your pain right now, on a scale from 0 to 10, where 0 is no pain and 10 is pain as bad as it could be?
► In the last 3 months, how would you rate your worst pain, on a scale from 0 to 10, where 0 is no pain and 10 is pain as bad as it could be?
► On average, in the last 3 months, how would you rate your pain on a scale from 0 to 10, where 0 is no pain and 10 is pain as bad as it could be?

The answers to these questions were then used to compute a variable on pain intensity on a scale of 0–4, indicating 'grade 0—no intensity (ie, no chronic pain)', 'grade 1—low intensity', 'grade 2—high intensity', grade 3—moderately limiting', 'grade 4—severely limiting'. This grading was based on the 3-item Graded Chronic Pain-Pain Catastrophizing Scale.[23] Sociodemographic variables included were age, sex, marital status, highest educational qualifications, occupational classifications, household income quintile. Health-related data included self-assessed general health status (very good, good, fair, bad, very bad), presence of mental health disorder (yes/no), anxiety levels (not anxious or depressed, moderately anxious or depressed, extremely anxious or depressed) and ranking of happiness on a 0–10 scale. Opioid usage data were also contained in the 2011 HSE. Area-level deprivation data included the Index of Multiple Deprivation (IMD) 2010 obtained from the HSE. The IMD produces a ranking of areas in England based on relative local scores for: income, employment, health, education, crime, access to services and living environment. IMD was included because there is a strong relationship between area-level deprivation and mortality and morbidity—with the most deprived neighbourhoods in England experiencing life expectancy 9 and 6 years less for men

and women, respectively, than those that are the least deprived.[24]

The English regions were classified as the North (North East, North West, Yorkshire and the Humber) and the South (London, East of England, West Midlands, East Midlands, South East and South West). This study used individual-level HSE data, and therefore, HSE survey weights applicable for individual-level data were used.

## Data analysis

Chronic pain prevalence, pain intensity and opioid usage were mapped using Adobe Illustrator with local and regional boundaries downloaded from the Office for National Statistics. In the HSE, opioid use was described as a binary variable (a yes or no response), and was used as an outcome variable to examine the association between opioid use and factors associated with it. The HSE 2011 individual-level data had 10617 cases, and pain data were only collected among respondents aged 16 years and over (n=8610). Cases where there were missing values for the confounding variables (regions, age, sex, marital status, highest educational qualification, occupational level, household income quintiles, general health status, mental health disorders, anxiety levels and happiness scale) were then excluded from our analysis. Missing values in the HSE can occur for several reasons, including refusal or inability to answer a particular question or refusal to cooperate in an entire section of the survey. After this, the dataset with no missing values (n=5711) was used in our analysis. Variables that showed significant bivariate association were included in the initial model. Apart from the presence of chronic pain and pain intensity, the initial model included age, sex, marital status, highest educational qualification, occupational level, household income quintiles, general health status, mental health disorders, anxiety levels and happiness scale. A generalised linear model with binomial distribution and logit link was used to examine the associations between opioid use and chronic pain, adjusted for individual-level and area-level covariates. Survey weight was applied to the model. The most parsimonious model was obtained by using likelihood ratio test statistics to ensure there was no significant loss of information. To support the spatial analysis of a 'pain divide' between the North and the South of England, the pain intensity data were analysed using a generalised logit model to simultaneously analyse the four logit models resulting from the five levels of the pain intensity data (no pain, low intensity, high intensity, moderately limiting and severely limiting). Although the pain intensity is ordinal, the proportional odds model is both intuitively and statistically not appropriate because of the assumption of the common odds between the levels of pain intensity data. Survey weight was used in all analyses to ensure generalisation of findings.

This study was undertaken and reported according to the Strengthening the Reporting of Observational Studies in Epidemiology (STROBE) recommendations.[25] Data analysis was done using SAS v.9.4.

## Patient and public involvement

As this study involved secondary data analysis from the HSE, patients or the public were not involved in the design or delivery of this research.

## RESULTS
### Regional inequalities in the prevalence of chronic pain, pain intensity and opioid use in England

The prevalence of chronic pain was 36.7% in the North of England, compared with the 35.0% in the South of England, as shown in table 1, and visually in figure 1. In terms of the nine English regions, the prevalence of chronic pain was highest in the North East, and lowest in London (43.1% vs 29.0%). In terms of pain intensity, 9.2% of people living in the South had 'moderately limiting' or 'severely limiting' chronic pain, while, in the North, 12.3% of people had 'moderately limiting' or 'severely limiting' chronic pain. People in the North were also more likely to experience 'moderately limiting' or 'severely limiting' pain than those in the South: the odds of severely limiting pain were 32% higher in the North than in the South; similarly, the odds of 'moderately limiting' pain were 37% higher in the North than the South, as shown in table 2. In addition to differing pain levels in the North and South English regions, there were also observed differences in anxiety and self-reported general health: anxiety levels in the North were 27.3%, compared with 25.7% in the South, while for self-reported general health, 7.6% and 5.5% of people living in the North and South, respectively, were reported to have 'bad' or 'very bad' health status. Although chronic pain prevalence was similar in the North and South of England (36.7% and 35.0%, respectively), opioid use was somewhat higher in the North (2.5%), compared with the South (1.7%). Furthermore, the use of opioids (weighted results) was higher in the North of England for people with 'severely limiting' chronic pain (16.9%), compared with people in the South (10.4%), as illustrated by figure 2.

### Association of opioid utilisation and chronic pain after adjusting for individual-level and area-level confounders

Opioid usage was significantly associated with chronic pain intensity (adjusted for age, household income, education level, general health and anxiety): in people with higher pain intensities, there were higher odds of opioid use, as illustrated by table 3. The use of opioids was also positively associated with household income levels: households belonging to the higher income quintiles had significantly higher odds of using opioids than those at the lowest quintile. In addition, general health status was significantly positively associated with opioid usage: people who reported 'very bad' or 'bad' health status had 14% higher odds, and 6% higher odds of using opioids, respectively, compared with those who reported 'very good' health status.

## DISCUSSION
This paper is the first to examine geographical inequalities in chronic pain prevalence, pain intensity and

**Table 1** Characteristics of the study population

| Variable | South % (n) | North % (n) | Overall % (n) |
|---|---|---|---|
| Region | 68.2 (3896) | 31.8 (1815) | 100.0 (5711) |
| Age group | | | |
| Median (25th, 75th percentile) | 45 (32, 60) | 45 (32, 60) | 45 (32, 60) |
| Sex | | | |
| Female | 55.9 (2178) | 55.3 (1003) | 55.7 (3181) |
| Male | 44.1 (1718) | 44.7 (812) | 44.3 (2530) |
| Marital status | | | |
| Single | 25.5 (995) | 27.7 (503) | 26.2 (1498) |
| Married/civil partner | 55.8 (2173) | 50.7 (921) | 54.2 (3094) |
| Divorced/widowed/separated | 18.7 (728) | 21.5 (391) | 19.6 (1119) |
| Anxiety grades | | | |
| Extreme | 2.2 (85) | 2.9 (53) | 2.4 (138) |
| Moderate | 23.5 (917) | 24.4 (443) | 23.8 (1360) |
| Not anxious | 74.3 (2894) | 72.7 (1319) | 73.8 (4213) |
| Income quintiles | | | |
| Highest | 25.1 (977) | 16.4 (297) | 22.3 (1274) |
| Second highest | 22.7 (884) | 20.2 (367) | 21.9 (1251) |
| Middle | 19.9 (775) | 20.6 (374) | 20.1 (1149) |
| Second lowest | 18.2 (708) | 25.1 (455) | 20.4 (1163) |
| Lowest | 14.2 (552) | 17.7 (322) | 15.3 (874) |
| Occupation | | | |
| Managerial and professional | 40.2 (1566) | 34.2 (620) | 38.3 (2186) |
| Intermediate | 25.2 (982) | 21.5 (390) | 24.0 (1372) |
| Routine and manual | 31.0 (1209) | 40.9 (743) | 34.2 (1952) |
| Other | 3.6 (139) | 3.4 (62) | 3.5 (201) |
| Educational qualifications | | | |
| No qualifications | 17.3 (674) | 21.8 (395) | 8.7 (1069) |
| Foreign/other | 1.4 (54) | 1.5 (27) | 1.4 (81) |
| NVQ level 1 or equivalent | 4.0 (155) | 4.7 (86) | 4.2 (241) |
| NVQ level 2 or equivalent | 22.2 (864) | 22.0 (399) | 22.1 (1263) |
| NVQ level 3 or equivalent | 15.6 (608) | 15.4 (279) | 15.5 (887) |
| NVQ level 4 or equivalent | 11.9 (462) | 12.2 (222) | 12.0 (684) |
| NVQ level 5 or equivalent | 27.7 (1079) | 22.4 (407) | 26.0 (1488) |
| General health | | | |
| Very bad | 1.3 (52) | 2.3 (42) | 1.6 (94) |
| Bad | 4.2 (165) | 5.3 (96) | 4.6 (261) |
| Fair | 15.6 (607) | 18.2 (330) | 16.4 (937) |
| Good | 44.1 (1719) | 42.8 (776) | 43.7 (2495) |
| Very good | 34.7 (1353) | 31.5 (571) | 33.7 (1924) |
| Mental health disorder | | | |
| No condition | 96.0 (3741) | 95.6 (1735) | 95.9 (5476) |
| Has condition | 4.0 (155) | 4.4 (80) | 4.1 (235) |
| Happiness scale | | | |
| Median (25th, 75th percentile) | 8 (7,9) | 8 (7,9) | 8 (7, 9) |

Continued

**Table 1** Continued

| Variable | South % (n) | North % (n) | Overall % (n) |
|---|---|---|---|
| **Weighted results with 95% CIs and numbers** | | | |
| Opioid use | | | |
| No | 98.3 (97.9 to 98.7; n=3897) | 97.5 (96.8 to 98.3; n=1577) | 98.1 (97.7 to 98.5; n=5474) |
| Yes | 1.7 (1.3 to 2.1; n=66) | 2.5 (1.7 to 3.3; n=40) | 1.9 (1.5 to 2.3; n=106) |
| Chronic pain intensity | | | |
| None | 65.0 (63.5 to 66.5; n=2574) | 63.3 (60.9 to 65.6; n=1024) | 64.5 (63.2 to 65.7; n=3598) |
| Low intensity | 1.7 (1.3 to 2.1; n=66) | 1.4 (0.8 to 1.9; n=22) | 1.6 (1.3 to 1.9; n=88) |
| High intensity | 24.2 (22.8 to 25.5; n=958) | 23.0 (21.0 to 25.1; n=373) | 23.8 (22.7 to 24.9; n=1331) |
| Moderately limiting | 3.4 (2.8 to 3.9; n=134) | 4.3 (3.3 to 5.3; n=69) | 3.6 (3.1 to 4.1; n=203) |
| Severely limiting | 5.8 (5.1 to 6.6; n=231) | 8.0 (6.7 to 9.4; n=130) | 6.5 (5.8 to 7.1; n=361) |

opioid utilisation in England. It is also the first to examine the association between chronic pain intensity and opioid utilisation. We have identified two key findings that may be of importance to healthcare practitioners and policy-makers: (1) there are geographical variations in chronic pain prevalence, pain intensity and opioid utilisation across the English regions—with evidence of a 'pain divide' with people in the North of England more likely to have higher intensity of pain; (2) opioid utilisation was significantly, and positively associated with pain intensity. The higher prevalence and intensity of pain in the Northern regions, as well

as a higher percentage of people belonging to lower education groups may only partly explain the higher rates of opioid usage found there. However, the number of people who used opioids in the survey was too small to support an interaction model between pain intensity and regions or a separate subgroup analysis for each region. These findings suggest the reason why people in the North East of England are prescribed more opioid analgesics than other parts of England is owing to the higher health need (pain). This is in keeping with wider studies of regional inequalities in health[16] and is a potentially important and significant finding given the

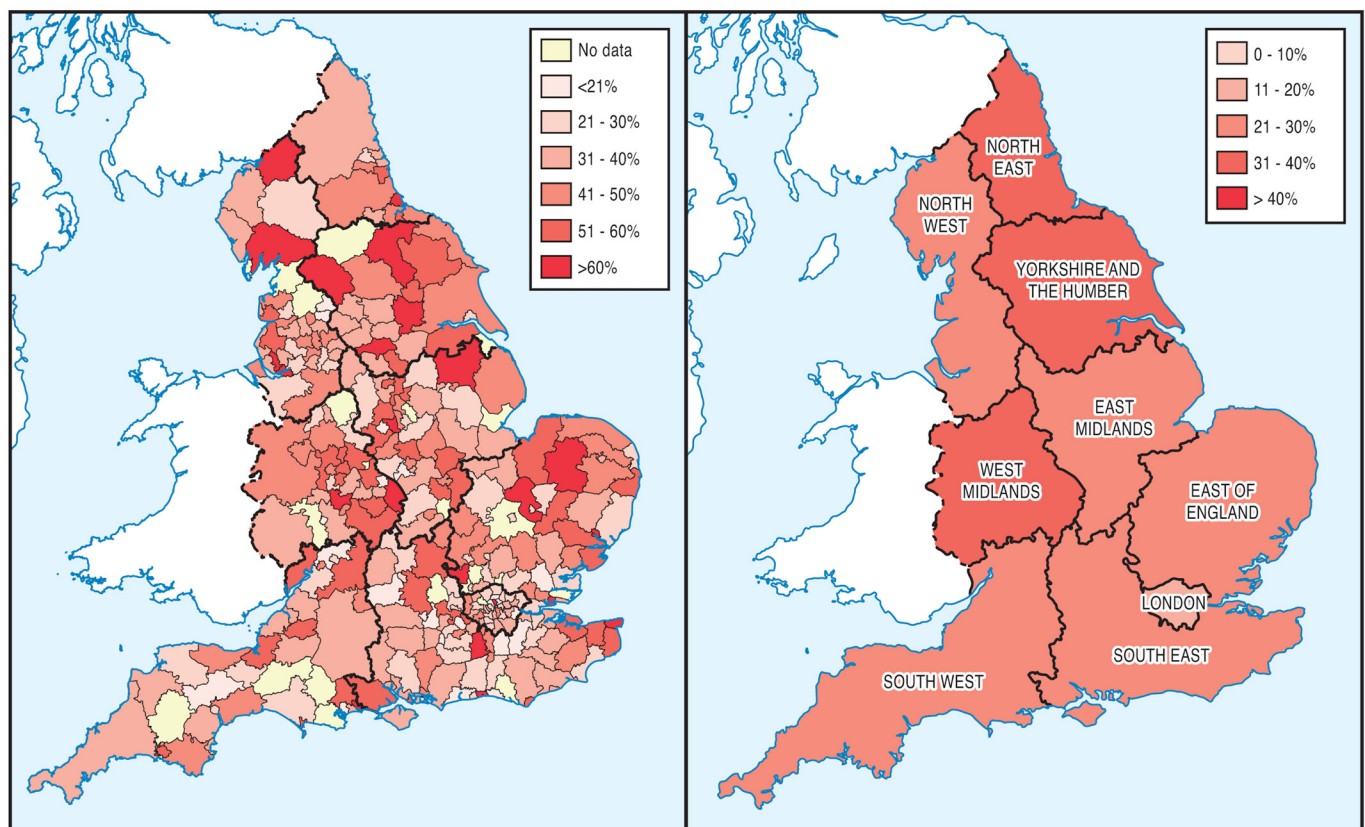

**Figure 1** Prevalence of chronic pain by local authority and English region.

**Table 2** Estimated ORs from generalised logit analysis of different pain intensities between the North and South of England adjusting for age, gender and level of qualifications.

| Variables | Categories | 'Severely limiting' versus 'no pain' | P values | 'Moderately limiting' versus 'no pain' | P values | 'High intensity' versus 'no pain' | P values | 'Low intensity' versus 'no pain' | P values |
|---|---|---|---|---|---|---|---|---|---|
| Intercept | | 0.011 (0.007, 0.018) | <0.001 | 0.010 (0.006, 0.018) | <0.001 | 0.097 (0.076, 0.124) | <0.001 | 0.004 (0.002, 0.010) | <0.001 |
| Region | North | 1.323 (1.063, 1.645) | 0.012 | 1.374 (1.035, 1.823) | 0.028 | 0.977 (0.852, 1.120) | 0.735 | 0.954 (0.604, 1.507) | 0.840 |
| | South | Ref | | Ref | | Ref | | Ref | |
| Age | | 1.042 (1.035, 1.050) | <0.001 | 1.033 (1.024, 1.042) | <0.001 | 1.030 (1.025, 1.034) | <0.001 | 1.036 (1.022, 1.054) | <0.001 |
| Gender | Female | 1.137 (1.020, 1.267) | 0.021 | 1.221 (1.059, 1.408) | 0.006 | 1.183 (1.108, 1.262) | <0.001 | 0.913 (0.738, 1.129) | 0.399 |
| | Male | Ref | | Ref | | Ref | | Ref | |
| Qualifications | None | 2.574 (2.069, 3.203) | <0.001 | 1.408 (1.021, 1.943) | 0.037 | 1.133 (0.964, 1.332) | 0.131 | 1.117 (0.660, 1.890) | 0.681 |
| | Foreign/other qualification | 1.188 (0.635, 2.222) | 0.591 | 0.873 (0.351, 2.171) | 0.770 | 0.894 (0.574, 1.392) | 0.619 | 1.373 (0.384, 4.911) | 0.626 |
| | NVQ level 1 or equivalent | 1.345 (0.872, 2.077) | 0.180 | 1.082 (0.587, 1.995) | 0.801 | 1.199 (0.917, 1.569) | 0.185 | 0.562 (0.164, 1.922) | 0.358 |
| | NVQ level 2 or equivalent | 1.057 (0.822, 1.358) | 0.667 | 1.073 (0.776, 1.483) | 0.669 | 1.007 (0.868, 1.169) | 0.927 | 0.748 (0.422, 1.326) | 0.320 |
| | NVQ level 3 or equivalent | 0.853 (0.621, 1.171) | 0.324 | 0.744 (0.487, 1.138) | 0.173 | 0.913 (0.768, 1.087) | 0.307 | 0.866 (0.459, 1.636) | 0.658 |
| | NVQ level 4 or equivalent | 0.718 (0.513, 1.005) | 0.054 | 0.937 (0.627, 1.401) | 0.752 | 0.999 (0.836, 1.194) | 0.991 | 1.162 (0.654, 2.065) | 0.609 |
| | NVQ level 5 or equivalent | Ref | | Ref | | Ref | | Ref | |

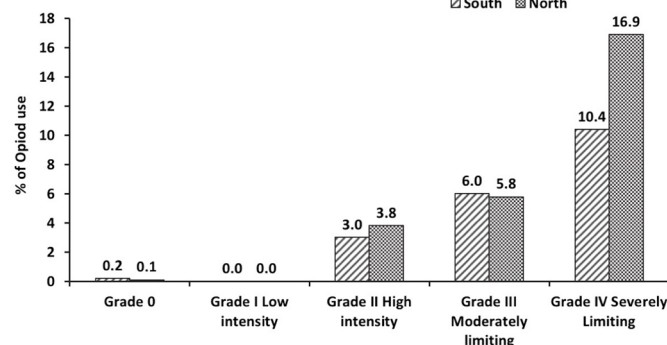

**Figure 2** Opioid use among participants from the North and South of England according to chronic pain grades.

recent public health concerns associated with opioid analgesics.

While this is the first study to examine the relationship between chronic pain intensity and opioid usage in England, there have been other studies that have explored the geographical variation in opioid prescribing. For example, a recent study by Mordecai *et al* showed that, at a clinical commissioning group level, over a 4-year period, there was an increasing trend of opioid prescribing—with more opioid analgesics prescribed in the North of England, compared with the South.[26] Our work builds on these findings, and shows that the increased trend of opioid prescribing is associated with an increase in health need (pain), rather than an 'inappropriate' prescribing trend of opioid analgesics. In addition to this, there have been a number of studies that have explored prescribing variation in other parts of the world, such the USA,[27 28] Canada[29] and Australia[30]; these studies have also showed there is a large geographical variation in prescribing practices of opioid analgesics, and call for guidance to promote good prescribing practices. Our results are timely, and show that, in England, the prescribing of opioid analgesics is largely driven by health need (pain); thus, to develop future strategies going forward, and avoid a potential 'opioid epidemic', as observed in the USA, it is important that consideration is given to other ways of managing chronic pain, without the use of opioid analgesics. While opioids may have a role in the short-term management of pain, their long-term use is questionable.[6–10] Currently, national guidelines recommend strong opioids as an option for pain relief for patients with chronic pain, providing they are reviewed annually, and only continued if they are providing ongoing pain relief.[31] While this is helpful in some instances, it is often difficult to ascertain, in a clinical setting, if opioid analgesics continue to provide ongoing pain relief; patients using opioids are also often reluctant to reduce or stop their opioid medication.[32 33] Studies also show that opioid discontinuation is associated with reducing pain scores; opioid-induced hyperalgesia also reduces on opioid cessation, which can further reduce levels of pain.[34] Given our findings, more needs to be done—at a national level—to support prescribers to manage people who have chronic

**Table 3** Generalised linear model examining associations between opioid use and chronic pain

| Variables | Categories | OR (95% CIs) | P values |
|---|---|---|---|
| Intercept | | 0.970 (0.956 to 0.985) | <0.001 |
| Age | | 1.000 (1.000 to 1.000) | 0.112 |
| Pain grade | Severely limiting | 1.078 (1.060 to 1.097) | <0.001 |
| | Moderately limiting | 1.036 (1.016 to 1.056) | <0.001 |
| | High intensity | 1.022 (1.013 to 1.031) | <0.001 |
| | Low intensity | 0.995 (0.968 to 1.023) | 0.746 |
| | No chronic pain | Ref | |
| Income quintile | Highest quintile | 1.017 (1.004 to 1.030) | 0.008 |
| | 4th | 1.018 (1.006 to 1.030) | 0.004 |
| | 3rd | 1.018 (1.006 to 1.030) | 0.003 |
| | 2nd | 1.016 (1.004 to 1.028) | 0.007 |
| | Lowest quintile | Ref | |
| Highest qualification | No qualification | 1.012 (1.000 to 1.024) | 0.059 |
| | Foreign/other | 1.005 (0.972 to 1.039) | 0.761 |
| | NVQ level 1 or equivalent | 1.002 (0.984 to 1.021) | 0.809 |
| | NVQ level 2 or equivalent | 1.004 (0.994 to 1.015) | 0.400 |
| | NVQ level 3 or equivalent | 1.011 (1.000 to 1.022) | 0.050 |
| | NVQ level 4 or equivalent | 1.005 (0.992 to 1.017) | 0.461 |
| | NVQ level 5 orequivalent | Ref | |
| General health | Very bad | 1.137 (1.102 to 1. 174) | <0.001 |
| | Bad | 1.057 (1.035 to 1.080) | <0.001 |
| | Fair | 1.022 (1.010 to 1.034) | <0.001 |
| | Good | 1.000 (0.992 to 1.008) | 0.995 |
| | Very good | Ref | |
| Anxiety | Extreme | 1.015 (0.991 to 1.039) | 0.227 |
| | Moderate | 1.008 (1.000 to 1.017) | 0.052 |
| | Not anxious | Ref | |

pain, without the need to initiate opioid analgesics. Another option that could potentially be used alongside this approach would be to consider how opioids are monitored and stopped in the community. We note the recent attention given to the term 'deprescribing'—a term used to describe the process of reducing or stopping inappropriate medication, with a view to minimising polypharmacy and improving patient outcomes.[35] It would be prudent to suggest that future prescribing strategies for opioids should also include an element of 'deprescribing' to ensure that if opioids are to be initiated, patients do not continue to use or be prescribed opioids for chronic pain indefinitely without benefit.

Our findings relating to geographical inequalities in chronic pain are in keeping with research into a number of other health outcomes, such as obesity, diabetes, cancer and cardiovascular disease, where higher rates are reported in the North—and in particular the North East—compared with the other English regions.[16] Our work suggests that the North South health divide could increase in the future unless prescribing practices change because current guidance for using opioids to manage pain means that the North will have a higher burden of adverse effects in the future. Further, with an ageing population and an associated increase in chronic conditions, then we anticipate a further increase in pain and therefore opioid use. Again, given the regional inequalities in the burden of disease, this could exacerbate further the North South divide. This is timely, as the recent Due North report,[18] an independent inquiry, commissioned by Public Health England, to identify actions that can reduce the gap in health between the North and South of England suggests that an urgent holistic approach is needed to ensure that future investment is effective at reducing inequalities. Our study shows that examination of the need for continued opioid prescribing should be considered in any strategies going forward to tackle the poorer health outcomes commonly reported in the North East of England, compared with the rest of the country.

In terms of study limitations, we acknowledge that there are several: first, in our analysis, we used chronic pain prevalence and pain intensity as the marker for

health need. Opioids are also used in the management of other conditions, such as acute postoperative pain, cancer pain or in the management of opioid substance dependence; clearly, this will have an influence regarding opioid prescribing practices. Also, the analysis does not discriminate between specific opioids, potency of opioid (eg, strong opioids vs weak opioids) or opioid dosages. It is also important to consider that geographical scale is important when exploring variation among a given area: it is possible that, even at local authority level, the opioid prevalence estimates are concealing further geographical patterning since they still contain relatively large populations. A finer scale analysis may, therefore, highlight particular opioid 'hotspots' where opioid prescribing and utilisation is concentrated. Another study limitation is that the HSE data were from 2011, although we note this is the most recent and meaningful data on chronic pain prevalence and pain intensity. Finally, the usual limitations of using cross-sectional data apply to this study meaning that we cannot claim causation only association. While we believe our results are robust, and have important policy implications, they should be interpreted cautiously in view of our acknowledged limitations.

## CONCLUSION

There are geographical differences in chronic pain prevalence, pain intensity and opioid utilisation across England—with evidence of a 'pain divide' with people in the North of England more likely to have 'severely limiting' or 'moderately limiting' chronic pain. In our model, the intensity of chronic pain was significantly, and positively associated with the use of opioid analgesics. Given the public health concerns associated with the long-term use of opioid analgesics—and their questionable activity in the management of chronic pain—more guidance is need to support prescribers in the management of long chronic pain, so the initiation of opioid can be avoided. Future opioid prescribing strategies should also consider incorporating deprescribing approaches to ensure when opioids are initiated, their use is regularly monitored, reviewed and discontinued in the community.

**Author affiliations**
[1]Institute of Health and Society, Faculty of Medical Sciences, Newcastle University, Newcastle upon Tyne, UK
[2]Fuse – the UKCRC Centre for Translational Research in Public Health, Newcastle upon Tyne, UK
[3]School of Pharmacy, Faculty of Medical Sciences, Newcastle University, Newcastle upon Tyne, UK
[4]Wolfson Research Institute for Health and Wellbeing, Durham University, Durham, UK
[5]School of Public Health Midwifery and Social Work, Canterbury Christchurch University, Canterbury, Durham, UK
[6]Department of Psychology, Durham University, Durham, UK
[7]Department of Biosciences, Durham University, Durham, UK
[8]Department of Pain and Anaesthesia, The James Cook University Hospital, Middlesbrough, UK

**Correction notice** Since this article was first published online, the acknowledgment statement has been moved to the funding section. The open access licence has changed from CC-BY-NC to CC-BY.

**Contributors** AT, CB and SE designed the study, and supervised all stages of the research. AT led the drafting of the manuscript with input from all authors. AK and NA led the statistical analyses; NW cleaned the data, conducted preliminary analyses and commented on the drafts. CB, AT and J-MC led on data interpretation. AE and PC informed the initial study design and commented on the analysis, and interpretation. AT is the corresponding author and acts as guarantor of the article.

**Funding** We would like to acknowledge the National Centre for Social Research for supplying us with the subnational HSE data required for the analysis. The Wolfson Research Institute for Health and Wellbeing supported us financially with a small grant to purchase the data. Chris Orton, Durham University, in cartography produced the map in this paper, and we thank him for his assistance. Author CB is a member of Fuse. Funding for Fuse comes from the British Heart Foundation, Cancer Research UK, Economic and Social Research Council, Medical Research Council, the National Institute for Health Research, under the auspices of the UK Clinical Research Collaboration, and is gratefully acknowledged (RF150334).

**Disclaimer** The views expressed in this paper do not necessarily represent those of the funders or UKCRC. The funders had no role in study design, data collection and analysis, decision to publish or preparation of the manuscript.

**Competing interests** None declared.

**Patient consent** Not required.

**Ethics approval** Ethical approval was not required for this work, as this study used non-patient identifiable secondary data; patients were not actively involved in this research.

**Provenance and peer review** Not commissioned; externally peer reviewed.

**Data sharing statement** Unfortunately, we are unable to share our data, as it does not belong to us. We have an agreement with HSCIC (source of our secondary data) that we will delete the data once we are finished using it.

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
