## [Reviewer comments · BMJ Open]

ARTICLE DETAILS

TITLE (PROVISIONAL)	The Pain Divide: A cross-sectional analysis of chronic pain prevalence, pain intensity, and opioid utilisation in England
AUTHORS	Todd, Adam; Akhter, Nasima; Cairns, Joanne; Kasim, Adetayo; Walton, Nick; Ellison, Amanda; Chazot, Paul; Eldabe, Sam; Bambra, Clare

VERSION 1 – REVIEW

REVIEWER	David Seamark University of Exeter Medical School, UK
REVIEW RETURNED	06-Jun-2017

GENERAL COMMENTS	I enjoyed reading this paper. The research question is important and I feel the team has done a good job in answering it. As always the results pose more questions than answers but the suggestions for further research made in the Discussion seem reasonable. It is important to emphasise that although opioid prescribing is increasing in England the situation is not directly comparable to that of the USA where it would appear that market pressures have contributed greatly to the 'opioid epidemic' in a radically different health care system. Not being an expert in modelling I found the graphics helpful in visualising the findings. Consequently it would be helpful to have a separate statistical review.
---

REVIEWER	Samantha Hollingworth School of Pharmacy University of Queensland Australia
REVIEW RETURNED	19-Jun-2017

GENERAL COMMENTS	This paper is a neat and important study linking four indicators of health concerned with pain and opioids at an ecological level in England. The paper would be improved by more attention to detail in writing (syntax and style), descriptions and tables. Introduction P5 line 10 define currency units e.g. US\$ P5 line 21 – adverse effects - also tolerance and dependence? P5 line 32 give population denominator for use and costs P5 line 34 talk about 'increased use' but did not give change over time – just for 2014 P5 line 36 Are there data on opioid misuse from the UK? P5 line 48. Costs of opioids – need population denominator. How are these regions defined? How many regions in England? Is the
--

	'northeast' a subset of the 'north'? What are the major cities in 'the north' (at least top five by population)? Remember, you will also have non-UK audience. Methods Clarify what opioids are measured in each metric. Are they consistent across the two measurements of prescribing and use? (also Discussion P16 line 10) P7 line 30 How did you calculate pain intensity? Provide more details or a citation P8 line 17 Are these all the regions in England. Figure 1 clarifies this but could do with some explanation in the text. P8 line ONS – cite reference Results P10 line 6 Table 1 – header row - Proportion of people (expressed as a percentage) P10 line 17 Did you consider a sensitivity analysis including West Midlands in the 'north'? It has high values. P10 line 31 give the risk estimate measure in the text - OR + 95% CIs? P10 line 42 – lower than what? 0.09 does not look like a large effect even if it is statistically significant P11 Table 2 likely no need for lower and upper CI in table row header. Suggest you keep the OR and 95%CI together in one column – no need for a separate 95%CI column – will also make it much easier to read. What is the 'constant? What are the reference categories e.g. for age? White ethnicity – compared to what other categories? Clearly show what region are in the south and north. Table needs revision. P13 line 54 whole para - refer to table/s in text as appropriate. The reference categories are not at all obvious. Table 3 How was % people prescribed opioids calculated? More details required e.g. if only had one prescription in year is that still counted? Population denominator? Table 4 – what are mean opioid 'estimates'? How was this calculated? Give some more details in methods. Consider using 'to' or a comma in 95%CI as '-' can be confused with a negative value. Discussion P15 line 17 Did you give an expected value that the observed values was 'over and above'? P16 line 25 You say "but are not reviewed or discontinued if the pain improves - and that this is particularly the case in some regions more than others" but provide no evidence for differential adherence- either from your data or from another study. P16 line 10. Is there any evidence that estimates OTC vs prescription opioid use in UK? References #12 consider a web link #16 A book? Usually provide a city #17 consider a web link #27 consider a web link
--	---

REVIEWER	Anthony Terrence O'Brien Neuromodulation center, USA
REVIEW RETURNED	04-Aug-2017

GENERAL COMMENTS	Summary Manuscript bmjopen-2017-017980 is cross-sectional study to quantify and analyze regional inequalities in chronic pain, pain intensity, opioid use and opioid prescribing in England. This study is unique to the region (though similar methodologies are published for other countries). It is also relevant considering the public health crisis around opioid use worldwide (particularly in the US). This study provides a “snap-shot” for appreciating prescription practices, planning future studies, and reviewing public health policies. The authors had 3 main objectives, to study: 1) if there were regional inequalities in prevalence and intensity of chronic pain in England; 2) if there were inequalities in regional opioid prescribing and use rates in England; 3) if inequalities in pain intensity explained regional inequalities in opioid prescription/use. To address these 3 objectives the authors used de-identified data from the NCSR- specifically they used data from HSE. This instrument included data on pain chronicity, intensity and opioid prescription. Relevant data were acquired from 2011 (a limitation recognized by the authors). Additional area level information was acquired from the Index of Multiple Deprivation (IMD) 2010, which was already incorporated in the HSE 2011, and had data for: income, employment, health, education, crime, access to services and living environment. Also, individual level covariates considered were age, sex, ethnicity and household income. Finally, area level opioid usage data were obtained from the NTA. Data were illustrated as maps for prevalence of opioid prescription, prevalence of chronic pain and for pain intensity by local authority and region in England. The authors describe the aforementioned in the text and via tables (they make descriptive comparisons of these data). Furthermore, a mixed multilevel model with the following dependent variables: chronic pain prevalence (dichotomized), pain intensity (continuous), opioid prescribing (dichotomized), and opioid usage (continuous) was constructed, where age, sex, ethnicity, household income, chronic pain, pain intensity, IMD and region were used as independent covariates. Key findings of this approach were that there were geographical variations in chronic pain prevalence, pain intensity, opioid prescribing and opioid utilization across the English regions. And that opioid prescribing and opioid utilization was not significantly associated with the prevalence of chronic pain. Indeed, the authors have done an exhaustive task to study a relevant public health topic, with a unique perspective on England. For this I commend the authors. Also, the manuscript is well written, and substantiated. Herein I point out a few suggestions for their consideration:
--

	Minor:  1. Some aspects of the “Data and variables” section and the “Data analysis” section are combined. For example on page 7 the authors refer to the HSE in the penultimate paragraph and then refer to the mixed multilevel model (MLM) as the MLM is part of the “Data analysis” section, I recommend that any mention of the model and any related methods be limited to the “Data analysis section”. 2. In the “Data analysis” section there should be a sentence on the descriptive analysis (which is presented later in results as percentages, pain scores etc.). 3. Number of subjects (denominator) included in each analysis should be mentioned. 4. For binary outcomes, presentation of both absolute and relative effect sizes is recommended (similar to previous). 5. Within the text there may be some confusion for the reader regarding the descriptive comparisons. For example, “For people experiencing chronic pain, their level of pain (i.e. pain intensity) was, on average, also higher in the North of England, compared to the South.”. As no inferential statistics were done for these singular comparisons (eg. t-test, ANOVA, Chi-squared) it may confuse the reader, and I suggest rewording the text (and similar sentences). 6. A table showing demographic and clinical characteristics may be considered useful. 7. To promote reproducibility, I suggest the authors describe any assumptions used to construct the MLM (eg. estimation method: Maximum likelihood vs. REML, if random effects model was used, what type of covariance structure was assumed?) 8. Variables used in the MLM were described as confounders. What is the bases of this assumption? Why couldn't they be significantly related covariates? The determination of these variables as confounders should be clear 9. In table 1 OR is used across the heading as the effect size. Is this correct, as only 2 of the models were based on logistic models. Were the beta co-efficients transformed for the linear models? 10. If it is possible to include colored illustrations it would help in the interpretation of the maps (which are exceptionally well done). Major:  1. How was missing data managed? 2. Like point 7 in minor issues, I suggest the authors explain the method used to determine the entry criteria for constructing the independent variables. For example in Vittinghoff E, Glidden DV, Shiboski SC, McCulloch CE. Regression Methods in Biostatistics: Linear, Logistic, Survival, and Repeated Measures Models. New York: Springer; 2005 $P \leq 0.2$ is used to minimize residual confounding. It is important that the method used to select the model is clearly stipulated as an exploratory approach generally is associated with high rates of false positive
--	---

	findings. 3. Following point 2, the approach for including/excluding variables into the model should be clearly described. Overall this is a very big task, and the authors have dedicated a lot of time and work to presenting vital information to the public in a concise and clear format. Additionally, they fully acknowledge limitations of the methodology. They have done a great job, contextualizing opioid usage in relation to chronic pain. I hope that any suggestions are taken with the best intentions. Please note I was specifically requested to focus on the statistical analysis used within the manuscript. Look forward to their responses. Respectfully. Peer-reviewer BMJ-Open
--	--

VERSION 1 – AUTHOR RESPONSE

Reviewer: 1

Reviewer Name: David Seamark

Institution and Country: University of Exeter Medical School, UK

Competing Interests: None declared

I enjoyed reading this paper. The research question is important and I feel the team has done a good job in answering it. As always the results pose more questions than answers but the suggestions for further research made in the Discussion seem reasonable.

It is important to emphasise that although opioid prescribing is increasing in England the situation is not directly comparable to that of the USA where it would appear that market pressures have contributed greatly to the 'opioid epidemic' in a radically different health care system.

Not being an expert in modelling I found the graphics helpful in visualising the findings. Consequently it would be helpful to have a separate statistical review.

Overall, I feel this paper should be accepted with minimal revision.

Thank you for the positive comments about our manuscript

Reviewer: 2

Reviewer Name: Samantha Hollingworth

Institution and Country: School of Pharmacy, University of Queensland, Australia

Competing Interests: None

This paper is a neat and important study linking four indicators of health concerned with pain and opioids at an ecological level in England.

Thank you for the positive comments – we have attempted to address your comments below.

The paper would be improved by more attention to detail in writing (syntax and style), descriptions and tables.

Introduction

P5 line 10 define currency units e.g. US\$

Done

P5 line 21 – adverse effects - also tolerance and dependence?

Yes we agree, and have amended the sentence accordingly

P5 line 32 give population denominator for use and costs

Done

P5 line 34 talk about 'increased use' but did not give change over time – just for 2014

The increased use is compared to 2004 – we have added a reference for the reader to consult, if they wish.

P5 line 36 Are there data on opioid misuse from the UK?

Nothing really that has been published as widely as the US; we have added an additional reference in a short communication that highlights the number of drug-related deaths from opioids in the UK (and that the number is increasing)

P5 line 48. Costs of opioids – need population denominator. How are these regions defined? How many regions in England? Is the 'northeast' a subset of the 'north'? What are the major cities in 'the north' (at least top five by population)? Remember, you will also have non-UK audience.

We agree with this, and many thanks for making the comment. There are nine regions in England – three of which are in the North (North East, North West and Yorkshire & Humber). We have amended our manuscript accordingly.

Methods

Clarify what opioids are measured in each metric. Are they consistent across the two measurements of prescribing and use? (also Discussion P16 line 10)

All opioids that fall under the British National Formulary (BNF) code 40702 (this covers all of the opioids currently used in clinical practice in the UK) – we cannot distinguish between opioids since they are all grouped together in the datasets unfortunately. We have made this clear in the text.

P7 line 30 How did you calculate pain intensity? Provide more details or a citation Pain intensity was already calculated in the HSE – we have provided a citation in text for more information on this.

P8 line 17 Are these all the regions in England. Figure 1 clarifies this but could do with some explanation in the text.

Yes – there are nine in total, which we have now included in the text.

P8 line ONS – cite reference

Thank you. We have now added this.

Results

P10 line 6 Table 1 – header row - Proportion of people (expressed as a percentage)

Changed.

P10 line 17 Did you consider a sensitivity analysis including West Midlands in the 'north'? It has high values.

We did not consider a sensitivity analysis and, unfortunately, do not have any capacity to do this at this stage, as our lead researcher has since left our group for another position.

P10 line 31 give the risk estimate measure in the text - OR + 95% CIs?

Done.

P10 line 42 – lower than what? 0.09 does not look like a large effect even if it is statistically significant This is lower than the North East of England. We acknowledge the difference is small, and as such, have added this caveat to the text (i.e. the difference is small, but yet significant)

P11 Table 2 likely no need for lower and upper CI in table row header. Suggest you keep the OR and 95%CI together in one column – no need for a separate 95%CI column – will also make it much easier to read.

We agree, and have amended the table.

What is the 'constant'?

We have omitted the term constant – it is what is used in the model but does not need to be presented here.

What are the reference categories e.g. for age?

There are no reference categories for age since it is a continuous variable.

White ethnicity – compared to what other categories?

All other categories – it is binary (White British, Irish & Scottish).

Clearly show what region are in the south and north. Table needs revision.

We haven't included which regions in North and South since the reference in the model is the North East and as such in this case it would be confusing to have sub-categories. We have, however, made it clear in the text what regions are in the North in response to an earlier comment.

P13 line 54 whole para - refer to table/s in text as appropriate. The reference categories are not at all obvious.

We have now added the reference to the table.

Table 3 How was % people prescribed opioids calculated? More details required e.g. if only had one prescription in year is that still counted? Population denominator?

As part of the nurse visits for the HSE, respondents were asked if they took any medication. This medication was recorded according to which British National Formulary category it fell into. We located the BNF code for opioids and the population denominator is the number of respondents in the HSE aged 15-64 years. Some text has been added to try to clarify this.

Table 4 – what are mean opioid 'estimates'? How was this calculated? Give some more details in methods.

It would take up too much space in our manuscript to go into this but we have provided a citation that explains the methodology in detail.

Consider using 'to' or a comma in 95%CI as '- ' can be confused with a negative value.

Good point – we have now changed this.

Discussion

P15 line 17 Did you give an expected value that the observed values was 'over and above'?

No – this was just a poorly worded phrase – we did not have an expected value. We have now removed the sentence from the manuscript to avoid confusion.

P16 line 25 You say "but are not reviewed or discontinued if the pain improves - and that this is particularly the case in some regions more than others" but provide no evidence for differential adherence- either from your data or from another study.

We acknowledge this, but in the previous sentence we do say that this is one of three potential reasons to explain our findings. We do agree with the comment about adherence, and have now added an additional sentence and reference.

P16 line 10. Is there any evidence that estimates OTC vs prescription opioid use in UK? No unfortunately not – only anecdotal – but this is something we would like to explore in the UK in future work.

References

#12 consider a web link Added
#16 A book? Usually provide a city Added
#17 consider a web link Added
#27 consider a web link Added

Reviewer: 3

Reviewer Name: Anthony Terrence O'Brien
Institution and Country: Neuromodulation center, USA
Competing Interests: None declared

Summary

Manuscript bmjopen-2017-017980 is cross-sectional study to quantify and analyze regional inequalities in chronic pain, pain intensity, opioid use and opioid prescribing in England. This study is unique to the region (though similar methodologies are published for other countries). It is also relevant considering the public health crisis around opioid use worldwide (particularly in the US). This study provides a “snap-shot” for appreciating prescription practices, planning future studies, and reviewing public health policies.

The authors had 3 main objectives, to study: 1) if there were regional inequalities in prevalence and intensity of chronic pain in England; 2) if there were inequalities in regional opioid prescribing and use rates in England; 3) if inequalities in pain intensity explained regional inequalities in opioid prescription/use.

To address these 3 objectives the authors used de-identified data from the NCSR- specifically they used data from HSE. This instrument included data on pain chronicity, intensity and opioid prescription. Relevant data were acquired from 2011 (a limitation recognized by the authors). Additional area level information was acquired from the Index of Multiple Deprivation (IMD) 2010, which was already incorporated in the HSE 2011, and had data for: income, employment, health, education, crime, access to services and living environment. Also, individual level covariates considered were age, sex, ethnicity and household income. Finally, area level opioid usage data were obtained from the NTA.

Data were illustrated as maps for prevalence of opioid prescription, prevalence of chronic pain and for pain intensity by local authority and region in England. The authors describe the aforementioned in the text and via tables (they make descriptive comparisons of these data). Furthermore, a mixed multilevel model with the following dependent variables: chronic pain prevalence (dichotomized), pain intensity (continuous), opioid prescribing (dichotomized), and opioid usage (continuous) was constructed, where age, sex, ethnicity, household income, chronic pain, pain intensity, IMD and region were used as independent covariates.

Key findings of this approach were that there were geographical variations in chronic pain prevalence,

pain intensity, opioid prescribing and opioid utilization across the English regions. And that opioid prescribing and opioid utilization was not significantly associated with the prevalence of chronic pain.

Indeed, the authors have done an exhaustive task to study a relevant public health topic, with a unique perspective on England. For this I commend the authors. Also, the manuscript is well written, and substantiated.

Thank you for the positive feedback.

Herein I point out a few suggestions for their consideration:

Minor:

1. Some aspects of the “Data and variables” section and the “Data analysis” section are combined. For example on page 7 the authors refer to the HSE in the penultimate paragraph and then refer to the mixed multilevel model (MLM) as the MLM is part of the “Data analysis” section, I recommend that any mention of the model and any related methods be limited to the “Data analysis section”.

Agreed – we have now moved this paragraph to data analysis section.

2. In the “Data analysis” section there should be a sentence on the descriptive analysis (which is presented later in results as percentages, pain scores etc.).

Thank you we have now added an additional sentence.

3. Number of subjects (denominator) included in each analysis should be mentioned.

We have now mentioned this in ‘Data and variables’ section and in Table 2.

4. For binary outcomes, presentation of both absolute and relative effect sizes is recommended (similar to previous).

Unfortunately, do not have any capacity to do this at this stage, as our lead researcher has since left our group for another position.

5. Within the text there may be some confusion for the reader regarding the descriptive comparisons. For example, “For people experiencing chronic pain, their level of pain (i.e. pain intensity) was, on average, also higher in the North of England, compared to the South.”. As no inferential statistics were done for these singular comparisons (eg. t-test, ANOVA, Chi-squared) it may confuse the reader, and I suggest rewording the text (and similar sentences).

Good point - we have now re-phrased to avoid confusion.

6. A table showing demographic and clinical characteristics may be considered useful.

Added (now Table 1).

7. I suggest the authors describe any assumptions used to construct the MLM (eg. estimation method: Maximum likelihood vs. REML, if random effects model was used, what type of covariance structure was assumed?)

We can confirm that no assumptions were made to construct the MLM.

8. Variables used in the MLM were described as confounders. What is the bases of this assumption? Why couldn't they be significantly related covariates? The determination of these variables as confounders should be clear.

We have used the wrong terminology here. They are significantly related covariates, not cofounders,

and as such that is why we have controlled for them in the analysis.

9. In table 1 OR is used across the heading as the effect size. Is this correct, as only 2 of the models were based on logistic models. Were the beta co-efficients transformed for the linear models? No, you are correct – this is an oversight. OR is only applicable to chronic pain and opioid prescribing which are binary variables. Pain intensity and opioid utilisation are scale/continuous variables and as such it should be labelled as a beta-coefficient. The table headers have been changed to make this clear and it has also been made clear in the text. Under the ‘Methods’ section we do have a sentence on this distinction ‘different multi-level models were used for each type of dependent outcome (logistic for binary and linear for scale/continuous)’.

10. If it is possible to include colored illustrations it would help in the interpretation of the maps (which are exceptionally well done).

Thank you. We have now included colour illustrations for the maps.

Major:

1. How was missing data managed?

The missing data was excluded. This has been added under ‘Methods’.

2. Like point 7 in minor issues, I suggest the authors explain the method used to determine the entry criteria for constructing the independent variables. For example in Vittinghoff E, Glidden DV, Shiboski SC, McCulloch CE. Regression Methods in Biostatistics: Linear, Logistic, Survival, and Repeated Measures Models. New York: Springer; 2005 $P \leq 0.2$ is used to minimize residual confounding. It is important that the method used to select the model is clearly stipulated as an exploratory approach generally is associated with high rates of false positive findings.

We checked correlation between the dependent and independent variables before including them in the models and they have been included in the analysis as they are significantly correlated with the outcomes under study ($p < 0.05$).

3. Following point 2, the approach for including/excluding variables into the model should be clearly described.

As above. A sentence has been added on this.

Overall this is a very big task, and the authors have dedicated a lot of time and work to presenting vital information to the public in a concise and clear format. Additionally, they fully acknowledge limitations of the methodology. They have done a great job, contextualizing opioid usage in relation to chronic pain. I hope that any suggestions are taken with the best intentions. Please note I was specifically requested to focus on the statistical analysis used within the manuscript. Look forward to their responses.

Thank you for your positive comments, and indeed with your review. We have certainly done our best to address your comments and feel the manuscript has improved as a result.

VERSION 2 – REVIEW

REVIEWER	Anthony Terrence O'Brien Neuromodulation lab, USA
REVIEW RETURNED	11-Sep-2017
GENERAL COMMENTS	The authors have clearly addressed all suggestions/questions. Minor recommendation to review tables with % and include the absolute number in the title or primary heading. Also for percentages authors

	may consider including the n used as 3/4 is not the same as 3000/4000 in absolute terms. Otherwise thank you for the opportunity to review your manuscript . Respectfully, Peer-reviewer BMJ-Open
REVIEWER	Samantha Hollingworth School of Pharmacy, The University of Queensland, Woolloongabba QLD, Australia
REVIEW RETURNED	14-Sep-2017
GENERAL COMMENTS	Thanks to the authors for a comprehensive response to all the issues.